# Geoparks and Education: UNESCO Global Geopark Villuercas-Ibores-Jara as a Case Study in Spain

**Rubén Fernández Álvarez**

Department of Geography, University School of Education and Tourism, University of Salamanca; 37008 Salamanca, Spain; rfa@usal.es; Tel.: +34-920-353-600 (ext. 3868)

**Abstract:** The UNESCO Global Geoparks is a good didactic instrument that allows teaching territorial elements in transversal way. The geopark provides a means to address the teaching-learning process of both cultural values and natural values, including geology. This paper aims to highlight the educational activities that have emerged from the Villuercas UNESCO Global Geopark. For this purpose, a qualitative methodology based on document analysis and fieldwork has been used. Through a system of categories, we have analyzed the existing materials on the geopark's website, its activity book and the schools that have been part of its educational project. The geopark and educational centers collaboration has 76 educational proposals that are developed in the territory of the geopark. Since 2011, all schools in its area of action participate in its educational project with a total of 2500 students per year. The Villuercas geopark involves all students in its territory and encourages learning of a significant, transversal and collaborative type. In the same way, it has a positive impact on environmental awareness and respect for the environment and cultural resources.

**Keywords:** UNESCO Global Geopark; geology; education; geography; Villuercas-Ibores-Jara

---

## 1. Introduction

In response to a LEADER European Cooperation Project, the year 2000 witnessed the launch of the European Geopark Network (EGN), the forerunner of today's UNESCO Global Geoparks [1]. It was a transnational project designed to socioeconomically galvanise and dynamise depressed rural areas [2]. The overall project initially consisted of four areas that are markedly different and unique in their own way: Parque Cultural del Maestrazgo (Spain), Réserve Géologique de Haute-Provence (Francia), Geopark Vulkaneifel (Germany), and the Petrified Forest of Lesvos (Greece) [3–5]. By 2001, the EGN already had 12 geoparks [6]. That same year, a cooperation agreement was signed between the EGN and UNESCO, which led to the implementation in 2004 of the Global Geopark Network (GGN) made up of 25 geoparks (17 in Europe and eight in China) [7]. The GGN now has 147 members across 41 countries. Four of these geoparks are cross-border, reflecting the network's integrating and collaborative nature [2], rendering it possible to develop geoparks beyond and across each nation's administrative boundaries by considering solely their geological, cultural and environmental values when delimiting them. The geoparks are part of the UNESCO International Geoscience and Geoparks Programme (IGGP) since 2015. This programme is designed to foster sustainable development, geoconservation and people's engagement in the drafting of land management and zoning plans [8]. The success of the public's contributions lies largely in people's prior instruction in the topics to be addressed [9].

The 2015 launch of the UNESCO Global Geoparks meant the definition of a new format of land development that focused its two mainstays of action on Geotourism and Education in environmental matters [10,11]. These are two key aspects for preserving the geological and geomorphological assets of the areas in which they are to be found and for fostering their population's socioeconomic development

in a sustainable manner with nature [12]. There are numerous studies that have highlighted the positive effect that geoparks have on the environment, such as the conservation of bio and geo-diversity and socioeconomic development. For example, in their study "The socio-economic benefits of establishing national geoparks in China", Xun and Ting [13] highlight the benefits geoparks have provided for local society and how this has had a positive knock-on effect on the environment. Along these same lines, McKeever et al. [14] and Farsani et al. [3]; have reported the strengthening of the local economy following the establishment of a geopark, which plays a dynamising role because it will constitute a resource for attracting tourists.

In step with the establishment of geoparks, geotourism emerged as a new concept in the 1990s [15], with the aim being to use tourism to raise awareness in matters related to geology and its conservation [16]. Hose [17] (p. 16) defines it as "The provision of interpretative and service facilities to enable tourists to acquire knowledge and understanding of the geology and geomorphology of a site (including its contribution to the development of the Earth sciences) beyond the level of a mere aesthetic appreciation". It is an open concept that can be developed [18]. Hose's conceptual approach [17] highlights two aspects: the infrastructures that pave the way for the development of tourism and the presence of geological and geomorphological features that set the area apart from others. This first appreciation [19] has given rise to other new ones; for example, in the United States it is considered another aspect of sustainable tourism, and it is seen as a way of contributing to the improvement of both the environment and society as a whole [20]. By contrast, in the United Kingdom and Australia, it is assigned more narrowly and almost exclusively to the ambit of geological tourism, as Ólafsdóttir and Tverijonaite [18] derive from the study by Joyce [21] that indicates that geology is the tourist offer's core component, and not like the preceding case in which the environmental context in which it took place prevailed. In the same vein as Joyce [21], Pásková [22] focuses geotourism on the presence of geological and aesthetic resource as the features of attraction, likewise indicating that it is a tourist offer in which geoconservation and learning are implicit.

In their attempt to harness the concept of geotourism, Ólafsdóttir and Tverijonaite [18] have conducted research involving a literature review of the materials published on geoparks, highlighting how the most accurate and widespread definition is the one made by Newsome and Dowling [12] in 2010. Both have set about defining geotourism by merging the proposals put forward by a series of scholars with the following outcome: "a form of natural area tourism that specifically focuses on geology and landscape. It promotes tourism to geosites and the conservation of geodiversity and an understanding of earth sciences through appreciation and learning. This is achieved through independent visits to geological features, use of geo-trails and viewpoints, guided tours, geo-activities and patronage of geosite visitor centres" (p. 4). It is a rounded and comprehensive definition that not only covers matters related to the tourist offer dealing with geology, but also includes geoconservation, landscape features, environmental awareness from the perspective of geology, and education [18,23].

Geotourism's evolution reflects the incorporation of new features that complement it and make it a type of tourism in which cultural aspects take pride of place [19,24–26]. This new way of approaching the tourist phenomenon has given rise to the notion of "geoeducation". This may act as a nexus of union between sustainable tourist initiatives and awareness in environmental matters, nurturing the transfer of knowledge from scientific circles to the local population, and from them to visitors [27].

Within the framework of these new initiatives geared towards sustainable socioeconomic development with the land, a new project is being launched whose main objective involves territorial development based on a unique geology. In 2011, the Villuercas-Ibores-Jara Geopark (Spain) met all the requirements for joining the European Geopark Network, and in 2015 it became a UNESCO Global Geoparks. Over these years, it has rolled out a robust operating project that is playing its part in the development of the local district. According to the specific dynamics of a geopark, Las Villuercas has two preferential lines of action: geotourism and geo-education.

This study's overriding purpose is to highlight the case of the UNESCO Global Geopark Villuercas-Ibores-Jara as a success story in the development of its educational project. At the same time,

it seeks to identify all the educational activities and teaching resources that the geopark provides as part of its project; reveal the number of students that have benefited from it, either through programmes organised by the geopark itself or through visits made to it; identify the stages of education targeted by the geopark; and describe the infrastructures it has for the implementation of its teaching project. This will involve a descriptive and interpretative analysis of the materials provided by the geopark through its website and those obtained through fieldwork at its facilities.

## 2. UNESCO Global Geoparks and Education: Geo-education

Geotourism is the main attraction in geoparks' economic dynamics, evolving and incorporating new features that complement it and convert it into a type of tourism in which cultural aspects are one more enticement [19,24–32]. For example, the study conducted by Mao et al. [33] aims to address the perception geotourists have and those aspects or facilities they expect. Their study concludes that these kinds of tourists mainly look for geology to broaden their knowledge and learning and understanding other cultures, as goals they aim to fulfil through their visit to a geopark [33]. This type of tourism of a specialised nature in which the broadening of knowledge plays a major role means that the educational facet, whether of a formal or informal nature, is of paramount importance [12,19,23,34–37].

Since the beginning of 2000, a type of tourism is being developed that is related to education, with increasingly more people travelling for the purpose of learning [38,39], to the extent that this is the main reason for the trip, relegating leisure and pleasure to second place [39]. Farsani et al. [34] consider that knowledge tourism has two preferential lines of action: scientific tourism and investigative tourism. The former is defined as "a travel topic grouping scientific attractions. It covers interests in visiting and exploring scientific landmarks, including museums, laboratories, observatories and universities" [34] (p. 3). In turn, and as regards investigative tourism [34], Farsani et al., based on Mieczkowski [40], describe it as a journey undertaken to initiate or conduct research processes. This new kind of tourism prompts the need for basic infrastructures that will enable visitors to cater for their demands in terms of either research or science. Thus, the demand made by visitors through which they express their desire to increase their knowledge [33] combines with one of the spheres of action of the condition of geopark (education in geology for raising public awareness about this territorial heritage) to give meaning to the term geo-education. This concept is increasingly growing in importance within the context of geoparks. This status reflects certain geological and environmental characteristics that render them unique and make them a focus of attraction for "education-seeking" tourists [28]. This involves a transfer of knowledge that leads to an increase both in environmental awareness [34,41–46] and in the instruction of the community living within the geoparks' sphere of influence [23] to enable people to understand what they need to do for their conservation and protection [47]. Geo-education may be seen as the instructive action that contains learning and knowledge transfer within the context of a geopark. This may be of an educational nature (in formal and informal teaching) or involving research.

Geoparks are providing an ever-greater number of educational schemes to further exploit their geological and geomorphological resources [46]. According to the study conducted by Farsani et al. [45], geoparks adapt their educational proposals to render them accessible to all kinds of users, ranging from specialised to everyday geological information. From the same perspective, Bobrowsky et al. [43] posit that geoscientists are essential for enabling society to understand the nature of local geological resources. Accordingly, if we merge the ideas contained in both studies [41,44], we find that the combination of geoparks and experts in the field may lead to a profound understanding of geology among the public at large if the right channels for the transfer of knowledge are created.

Geoparks are furthermore understood to be assets that contribute to the economic diversification of the areas where they are located, as spaces for generating knowledge and raising the public's environmental awareness [2,19,26,31,45,48,49]. Some of their foremost lines of action, therefore, should be those focusing on research collaborations with universities and research centres, like those in place with schools. In order to cater for this latter aspect, geoparks need to develop a robust educational project that encapsulates the didactic schemes they are going to put in place, as well as have the

appropriate infrastructures and staff to enable them to proceed as the kernel for the transmission of knowledge [47]. Although these means may make self-learning possible, geoparks have lines of action within the scope of formal education, forging close ties between them and schools. This opens the door to knowledge on geology through basic education and creates a society with a certain degree of awareness, not only in matters of geology but also regarding the environment [18,33,40,44,50]. Furthermore, the people living on geoparks need to have a profound understanding of the local area and its resources in order to provide visitors with a better tourist experience.

## 3. UNESCO Global Geopark Villuercas-Ibores-Jara: Geographical and Geological Settings

UNESCO Global Geopark Villuercas-Ibores-Jara, established in 2011, is located in the southeast quadrant of the Spanish province of Cáceres, in the Autonomous Community of Extremadura, bordering to the south with the province of Badajoz, which is also in Extremadura, and to the east with Toledo, which is in the Autonomous Community of Castilla-La Mancha (see Figure 1 and see Table 1). It is a mountainous area that in morphological terms is part of the Montes de Toledo range. It occupies the central part of the Hercynian arc that links the Montes de Toledo with another range called the Sierra de Monfragüe. In stratigraphic terms, it belongs to the "*complejo metamórfico extremeño que se encuentra integrado por materiales precámbricos y paleozoicos*" (*metamorphic complex of Extremadura that consists of pre-Cambrian and Palaeozoic materials*) [51] (p. 97) (see Figure 2 and see Table 1).

It consists of a folded relief that has been worn down by the erosive processes of the Mesozoic and rejuvenated during the Alpine orogeny [51–53]. It consists of a stratigraphic network that is characterised by the alternation of hard materials (Armorican quartzite) and soft ones (slates) [54] (see Figure 2). This, together with differential erosive processes, has formed an Appalachian system in which there is a succession of iso-altitudinal ridge crests with deep valleys that run mainly in a NW-SE direction [51,54,55] (see Figure 3).

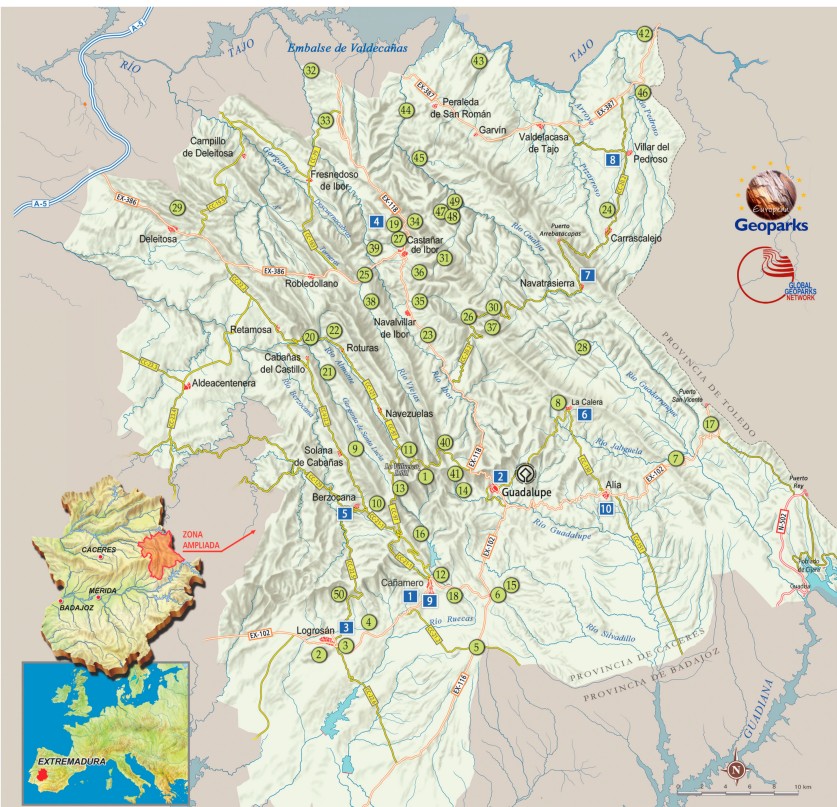

**Figure 1.** Location of the UNESCO Global Geopark Villuercas-Ibores-Jara and distribution of geosites. Source: UNESCO Global Geopark Villuercas-Ibores-Jara [56].

**Table 1.** Geoparks´Geosites (Figure 1). Source: UNESCO Global Geopark Villuercas-Ibores-Jara [56].

| Geosites | |
|---|---|
| 1 Crag of La Villuerca | 26 Peat bog of El Hospital del Obispo |
| 2 San Cristobal Hill | 27 Castañar Cave |
| 3 Costanaza Mine | 28 Crags of La Trucha |
| 4 Serradillo Mine | 29 Fracture system of La Breña ridge |
| 5 Fanglomerates of Cañamero and Alía | 30 Crags of El Ataque |
| 6 Fault of Puertollano | 31 Waterfalls of Calabazas |
| 7 Gorge of Peña Amarilla | 32 Ibor Potholes |
| 8 Limestone Klins of La Calera | 33 Metamorphic contact |
| 9 Crag of El Reloj | 34 Castañar Black Slopes |
| 10 Crags of Las Sábanas | 35 Magnesites of Navalvillar |
| 11 Source of the river Almonte | 36 Castañar Hill |
| 12 Gorge of the river Ruecas | 37 Carbonero Crag |
| 13 El Brazo mountain pass. Watershed slopes | 38 River Viejas-Torneros Valley syncline |
| 14 Guadalupe Valley and Valdegracia Ravine | 39 Hill of the mine. Clouding site, Ibor river |
| 15 Las Amoladeras | 40 Sebelidites y Vendotenidos site |
| 16 la Madrila Ridge | 41 Arca del Agua watermines |
| 17 Guadarranque Syncline | 42 Ravine of El Pedroso |
| 18 Gordo Mountain rock and Pimpollar ridge | 43 Spheroidal weathering of Peraleda |
| 19 Fanglomerates of Las Mesillas | 44 River Gualija Valley and Marialina Mine |
| 20 Rock pass of the river Almonte | 45 Crags of El Vadillo |
| 21 Santa Lucía Syncline | 46 Olistastrome of El Pedroso |
| 22 River Almonte Anticline | 47 Diamictites of El Endrinal Gorge |
| 23 River Ibor-Guadalupe Anticline | 48 Porrinas Threshing-floor. Contact Metamorphism |
| 24 Valdelacasa Anticline | 49 Paper Houses. Gualija river meanders |
| 25 Fault mirror and Crag of Las Narices | 50 Cabeza del Agua surroundings |

In terms of climate, the prevailing conditions are temperate with a hot, dry summer (Csa, Csa and Csb = Sub-humid temperate climates with hot summers according to the Köppen climate classification), in which drought is one of its most defining features. In the higher zones, there is a transition to a more temperate climate with a warm dry summer (Csb), although there is not a clear difference that informs a real change in climate. The Villuercas Geopark features a medium-sized Mediterranean mountain landscape characterised by the alternation of rocky ridge crests and valleys occupied by extensive livestock concerns in their low and medium sectors. These livestock usages have created the pasturelands that alternate with olive groves spread across the flat scrublands in the south and centre-north as the area's most typical exploitations.

The geopark has a surface area of 2544.4 km$^2$ spread across 19 local councils. It has a population of 12,883 people (January 2019), of whom 6573 are males and 6311 are females. The population's ageing index is 388.4% (IV = (P ≥ 65/P ≤ 15) × 100), with an average of 51.5 years. These two figures provide a general snapshot of the ageing process in Las Villuercas. In turn, the average population for the geopark as a whole is 5.06 inhabitants/km$^2$, which places this area below the threshold of a demographic desert (10 inhabitants/km$^2$). It is an area in which ageing and the loss of population are the prevailing demographic aspects, with depopulation posing a very high risk [57]. The population aged 65 and over accounts for 33.2% of the total, compared to 8.5% of the population aged under

16. Since the 1950s, and as a result of the rural exodus, the population has fallen constantly, and increasingly so in the last twenty years (see Figure 4).

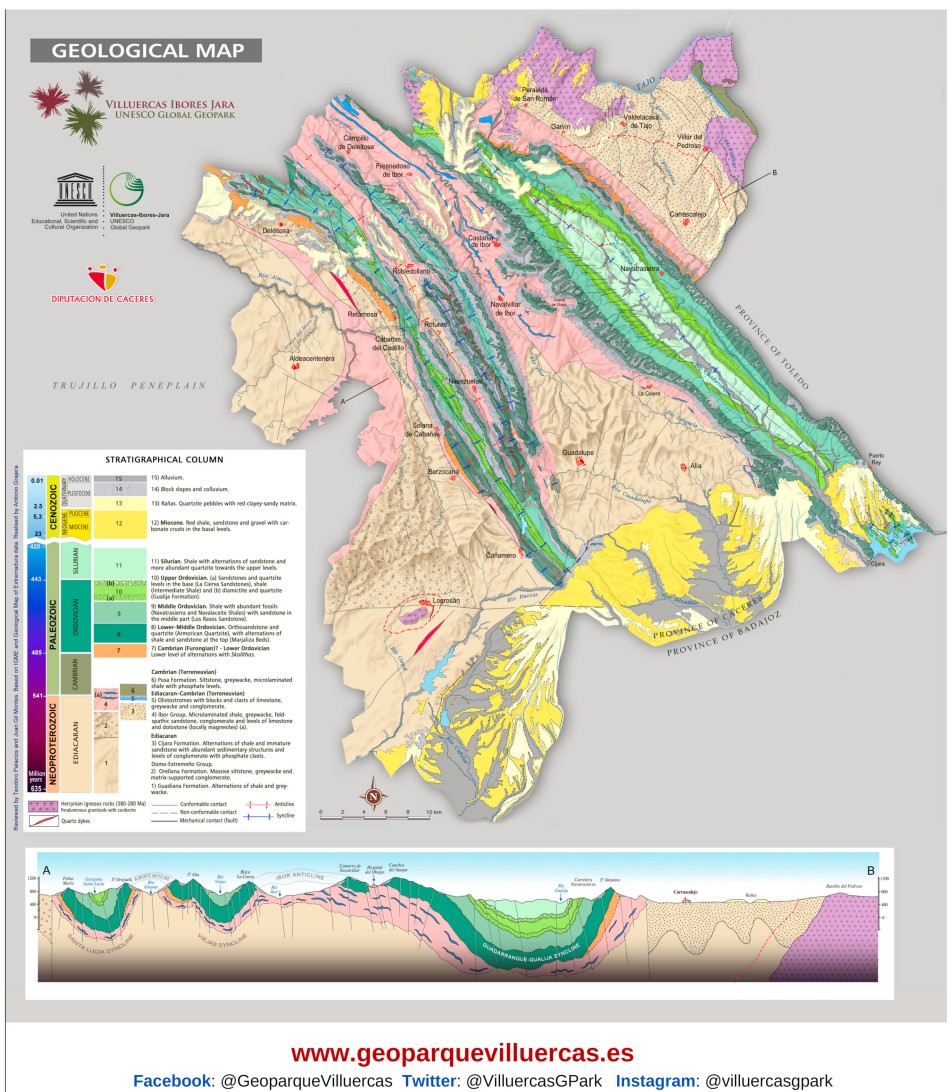

**Figure 2.** Geological map of the Villuercas. UNESCO Global Geopark Villuercas-Ibores-Jara [56].

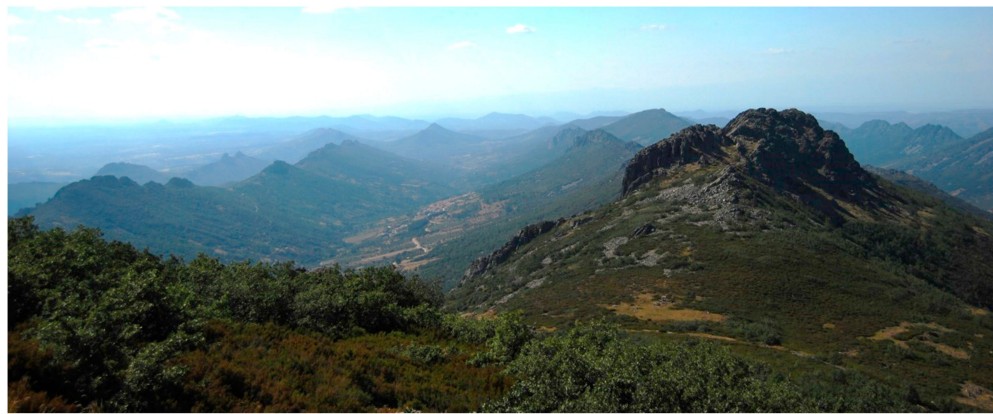

**Figure 3.** Iso-altitudinal ridge crests. Source: author's own work.

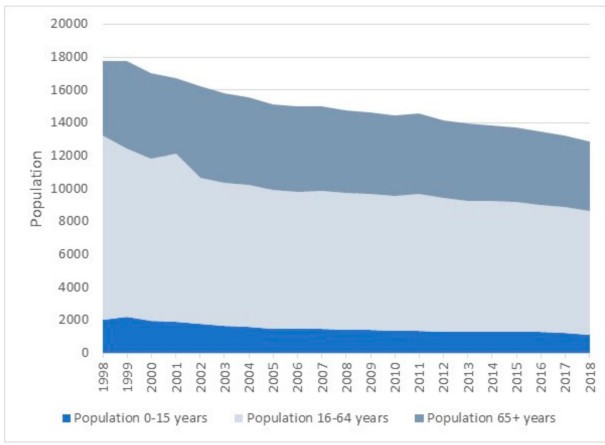

**Figure 4.** Trend in the population of the Villuercas-Ibores-Jara Geopark since 1998. Source: based on data provided by Spain's National Office of Statistics—Instituto Nacional de Estadística.

These population imbalances, in which the loss of population and ageing are the dominant trends, have a direct impact on the dependency rate (71.5%) (TA = [P > 65 + P ≤ 15/P (16 − 64)] × 100). This figure indicates that for every 100 individuals of working age in the district of Las Villuercas there are 71.5 dependents, of whom most have reached retirement age.

These basic demographic parameters reveal that the population of Las Villuercas is in a perilous state. Generational renewal is at risk and the trends indicates that depopulation is one of its main problems. The geopark is therefore a new feature that can help to mitigate the current situation through economic diversification. Accordingly, geotourism and its associated activities are some of the local resources that can boost the area's economy and provide incentives for attracting a new resident population.

## 4. Materials and Method

In terms of methodology, this article is structured around two main pillars: fieldwork and the analysis and interpretation of documents. The fieldwork sessions were planned with a view to analysing the infrastructures available in the geopark. Thisprovided access to visitor and interpretation centres, geosites, and georoutes. These sessions took place over four days (May 2019) that involved identifying those features available for teaching–learning processes and for geotourism on each georoute and at each geosite. The second part involved a qualitative and interpretative methodology based on the technique of analysing documents [58–60]. It was a method for the induction of thematic categories [60] that was undertaken from the explicit references located in the documents analysed. This process involved MAXQDA 18.2 software (is a product of VERBI GmdH, Germany), which was used to find in the text and analyse a series of thematic categories that allowed defining the educational activities held in the geopark according to their more representative features, as well as their methodological aspects and the materials used.

Following an initial process of analysing the documents to outline the activities (characteristics, methodology, objectives, etc.), five major thematic categories were identified: territorial elements; geology; human elements; awareness actions; and education (materials and methods involved in the activities). All these are consistent with the educational postulates underpinning geoparks and abide by the general principles of the European Geoparks Charter: geo-knowledge; environmental awareness; and cultural and territorial knowledge. Each one of these categories involves keywords that enable the accurate identification of the characteristics of the documentation and activities (see Table 2). Moreover, they will also lead to the identification of the secondary topics that are incorporated into the activities on a mainstream basis. The subcategories have been extracted from the analysis of the texts and the web in order to be able to recognize the main and secondary orientation of the educational activities.

**Table 2.** Categories and keyword. Source: author's own work.

| Thematic Category | Thematic Subcategory |
| --- | --- |
| Territorial elements | Geopark |
| | Territory |
| | Landscape |
| | Environment |
| | Nature |
| | Geography |
| | Geotourism |
| Geology | Geology |
| | Relief |
| | Fossil |
| | Mineral |
| | Rock |
| | Geosites |
| | Appalachian |
| Education | Play |
| | Geosuitcase |
| | Outing |
| | Teaching |
| | Learning |
| | Education |
| | Early Childhood |
| | Primary |
| | Compulsory Secondary |
| | Sixth Form |
| | Knowledge |
| | Science |
| | Transference |
| | Research |
| Awareness actions | Preservation |
| | Care |
| | Respect |
| | Awareness |
| | Conservation |
| | Participation |
| Human elements | Heritage |
| | Uses |
| | Culture |
| | Social |
| | Mine |

The following materials have been used for analysing the documents: *Geoexperiencias. Libro de actividades del Geoparque Villuercas-Ibores-Jara*; educational sections on the website of the UNESCO Global Geopark Villuercas-Ibores-Jara; map of geosites and descriptive documents; map of georoutes and descriptive materials [56,61].

## 5. Results

### 5.1. The Geopark's Educational and Tourist Infrastructures

The UNESCO Global Geopark Villuercas-Ibores-Jara has a series of facilities available for organising geotourist and geo-educational activities. A total of 50 geosites have been located and identified that showcase some of the more representative characteristics of the area's geological evolution [62] (see Figure 1). They all have qualities of a scientific and educational nature, in addition to a markedly aesthetic interest [61]. Although most of them are natural phenomena, it has five that have been developed to showcase the traditional mining operations that took place in the past.

For each one of these geosites, the geopark has issued a briefing document that explains the main aspects, as well as their location and access [62]. In addition, ten routes have been mapped out to enable visitors to go on self-guided walks following the landmarks or signs that mark the way. What's more, five of these routes cater for groups supervised and instructed by one of the geopark's dedicated guides [62], providing visitors with 95.2 km of scientific-educational routes. These geological resources (geosites and georoutes) are complemented by ten facilities (six interpretation centres, two museums, a mine, and a visitor centre) [62]. These provide tourists and students with information on the geopark and its resources, as well as the opportunity to sign up for workshops and cultural activities.

In addition to this, the geopark has published a catalogue of tourist services that lists all the geotourist activities organised on the geopark by public and private institutions. This catalogue describes 21 tourist initiatives, as well as the catering and hospitality options available [56].

### 5.2. Geoschool Educational Project: Description and Features of Its Activities

Since 2011, the UNESCO Global Geopark Villuercas-Ibores-Jara has been developing its own educational project called "Geocentros" (Geoschools). It involves a series of schemes based on the teaching of geology and knowledge of our cultural heritage designed to raise cultural awareness. It is a holistic educational project in which the geopark has been used as the focal point; it has been turned into a classroom in the midst of nature. This project involves all the primary and secondary schools within the park's area of influence. The aim is to improve the teaching-learning process and include fieldwork as a permanent educational resource. In methodological terms, this project has the following features: participative; engaging; constructivist; and mainstream. All the didactic propositions are based on motivation, and their development involves the steady incorporation of content to enable students to build their own learning process in a responsible manner [61]. Furthermore, all the activities are divided into two parts: an initial theoretical study followed by practical work. Although it is designed mainly for schoolchildren, the project can also be extended to everyone living in the vicinity of the geopark. The geopark's management considers it vital to further knowledge so that everyone living in the area can appreciate their local wealth [61]. The aim is therefore to ensure that local people become the guardians of their own environment [61].

The following are the mainstays of the Geoschools project: The engagement of society at large; a flexible approach with the ability to adapt to social expectations; creating a thinking society; the implementation of viable educational schemes; fostering social participation; including educational processes in all schemes and initiatives; boosting collaboration; and providing the necessary resources [61]. This requires the collaboration and cooperation of all stakeholders, whether these are social, educational or public bodies (local authorities, the Provincial Council or Diputación of Cáceres, and the Regional Government, the Junta, of Extremadura). This necessary coordination has involved the creation of a task force whose members are delegates representing each stakeholder group involved,

with the remit to define a viable and accessible educational strategy for all the schools and for society as a whole. Within the ambit of compulsory schooling, tutors will therefore be the ones responsible for defining the specific actions to be undertaken in their classrooms.

The Geoschools project has a knowledge transfer programme through the Google+ community. It is a channel for sharing experiences and materials among schools and teaching staff. In addition, it acts as a focal point for the coordination and planning of some of the educational activities. This means that those activities of a global nature for the sum of schools can be efficiently managed.

5.2.1. Description and Characterisation of the Geopark's Educational Activities and Materials.

Over the years the Geoschools project has been in operation, the Villuercas-Ibores-Jara Geopark has produced a series of educational proposals and materials for learning about geology, the environment, and its cultural heritage (see Tables 3 and 4). Their aim is to reinforce the geopark's value as a resource for the teaching-learning process and raise people's awareness about the protection and sustainable use of local resources. Both the materials and the teaching proposals are designed and created for the levels of primary and secondary educations, adapting to the children's needs and levels. This means covering two stages of compulsory schooling (primary education and compulsory secondary education) to reach out to all pupils, as well as to the prior and subsequent stages (early childhood and the sixth form). In terms of early childhood education, the aim is to provide play activities in the countryside that, through this first contact, teach the children the values of respect for nature and its conservation. In the two years leading up to university (*bachillerato* or sixth form), the students acquire specific knowledge of geology and the conservation of biodiversity.

**Table 3.** Teaching materials created by the Villuercas-Ibores-Jara Geopark. Source: UNESCO Global Geopark Villuercas-Ibores-Jara and Geopark Text-book. [56,61].

| Teaching Resources | Educational Stage |
|---|---|
| Field dossier | Early Childhood, Primary and Compulsory Secondary |
| Didactic sheet | Early Childhood, Primary and Compulsory Secondary |
| Geosuitcase | Early Childhood, Primary and Compulsory Secondary |
| Textbook in the subject Natural Sciences in Villuercas Geopark | Primary and Compulsory Secondary |
| Geoexperiences book | Early Childhood, Primary and Compulsory Secondary |
| Time wall (Fausto Maldonado School) | Early Childhood and Primary |
| Wildlife wall (Fausto Maldonado School) | Early Childhood and Primary |
| The rock-climbing of geological time (Licinio de la Fuente School) | Early Childhood and Primary |

The teaching materials drafted as part of this educational project follow a common denominator in which knowledge of geology and geoparks are the main factors. They have been created for use in several stages of schooling, and teachers can adapt them to their pupils' level. Six of the eight materials have been designed for classroom use or in the actual schools. These involve such highlights as the geosuitcase and the textbook. The former is a supplementary resource in the process of teaching-learning geology. It consists of a box containing a series of minerals and fossils that are used as examples in the process. The textbook is designed to complement the official textbook in the subject natural sciences in primary education and in the first two years of secondary. It provides a new instrument for working on the subject, with Las Villuercas Geopark as its common thread. The material to be used on excursions consists of a field dossier with activities, supplementary information, and space for notes. Both the rock-climbing wall and the other two walls (geological time and wildlife) constitute the so-called "edu-sites". The Edu-sites are new points of interest in the geopark like the geosites, whose school educational purpose is fundamental (see Figure 5).

**Table 4.** Teaching activities held at the schools located in the geopark. Source: UNESCO Global Geopark Villuercas-Ibores-Jara and Geopark Text-book. [56,61].

| Didactic Activity | Educational Level | Organized by |
|---|---|---|
| Geocoexistence (Geoconvivencia) | Early childhood, primary and compulsory secondary | Geopark |
| Educational workshops | Early childhood, primary and compulsory secondary | Geopark |
| European Geoparks Charter | Primary and compulsory secondary | Geopark |
| Geopark tour | Early childhood, primary and compulsory secondary | Geopark |
| I´m a Geoparker | Primary and compulsory secondary (informal teaching) | Geopark |
| School newspaper "Amanecer" | Early childhood and primary | School (CEIP [1] Licinio de la Fuente) |
| The animals of my environment | Early childhood | School (CEIP Licinio de la Fuente) |
| Geovalues (Ethical values) | Early childhood and primary | School (CEIP Licinio de la Fuente) |
| Trekking | Early childhood, primary and compulsory secondary | School (CRA [2] Montellano; IESO [3] Las Villuercas) |
| Excursion to the caves | Early childhood and primary | School (CRA Montellano) |
| Life in Prehistory | Early childhood | School (CRA Montellano; CEIP Fausto Maldonado) |
| History and art | Primary | School (CRA Montellano) |
| Ecological orchard | Early childhood and primary | School (CRA Montellano; CEIP Fausto Maldonado) |
| Knowing professions (chees-maker) | Early childhood and primary | School (CRA Montellano) |
| Knowing professions (baker) | Early childhood and primary | School (CRA Montellano) |
| Cave painting workshop | Early childhood and primary | School (CRA Montellano; CEIP Fausto Maldonado) |
| Geopark words | Primary | School (CRA Montellano) |
| Excursion to Logrosán | Early childhood and primary | School (CRA Montellano) |
| The slate | Early childhood | School (CEIP Fausto Maldonado) |
| Old crafts in The Villuercas | Primary | School (CEIP Fausto Maldonado) |
| Knowing the geopark rocks | Primary and compulsory secondary | School and High School (CEIP Fausto Maldonado; IES [4] Mario Roso de Luna) |
| Knowing the geopark fossils | Primary | School (CEIP Fausto Maldonado) |
| The bees and the geopark | Primary | School (CEIP Fausto Maldonado) |
| Bee products | Primary | School (CEIP Fausto Maldonado) |
| Study of the relief and geography of Cañamero | Primary | School (CEIP Fausto Maldonado) |

**Table 4.** *Cont.*

| Didactic Activity | Educational Level | Organized by |
|---|---|---|
| Children's games with rocks | Primary | School (CEIP Fausto Maldonado) |
| Experiences with the Californian red earthworm | Primary | School (CEIP Fausto Maldonado) |
| Geopark alphabet soup | Primary | School (CEIP Fausto Maldonado) |
| Tree leaf classification | Early childhood | School (CEIP Fausto Maldonado) |
| Eating chestnuts | Early childhood and primary | School (CEIP Fausto Maldonado) |
| Rock paintings mural | Early childhood and primary | School (CEIP Fausto Maldonado) |
| Rock paintings T-shirt | Early childhood and primary | School (CEIP Fausto Maldonado) |
| Rock exhibition | Primary | School (CEIP Fausto Maldonado) |
| Lighting in the mine | Primary | School (CEIP Fausto Maldonado) |
| Plant workshop | Primary | School (CEIP Fausto Maldonado) |
| Drawing the geopark | Primary | School (CEIP Fausto Maldonado) |
| Once upon a time | Primary | School (CEIP Rodrigo Dávila Martín) |
| Geoplayground project | Early childhood and primary | School (CEIP Rodrigo Dávila Martín) |
| Exhibition "How was the geopark" | Early childhood and primary | School (CRA La Jara) |
| Craft elaboration | Primary | School (CRA Las Villuercas) |
| Model of the mountains of the Villuercas | Primary | School (CRA Las Villuercas) |
| Drawing nature | Primary | School (CRA Las Villuercas) |
| Herbarium | Primary | School (CRA Las Villuercas; CEIP Fausto Maldonado; CRA La Jara) |
| Bird Feather Mural | Primary | School (CRA Las Villuercas) |
| Baby wild pig | Primary | School (CRA Las Villuercas) |
| Comic "The Villuercas" | Primary | School (CRA Las Villuercas) |
| Route through the Breña | Early childhood and primary | School (CRA Las Villuercas) |
| Comic "Walking Through the Villuercas" | Early childhood and primary | School (CRA Las Villuercas) |
| Looked at the geopark | Primary | School (CRA Las Villuercas) |
| A living geopark in Las Villuercas | Early childhood and primary | School (CRA Las Villuercas) |
| Our pets in spring | Primary | School (CRA Las Villuercas) |
| The Costanaza Mine | Primary | School (CEIP Nuestra Señora del Consuelo; CRA Montellano) |
| Geopark collage | Primary | School (CEIP Nuestra Señora del Consuelo) |
| Geopark geosites | Adult education | Centro escolar (EPA [5] Logrosán) |
| Solar energy workshops | Compulsory secondary | School (IESO Las Villuercas) |

**Table 4.** *Cont.*

| Didactic Activity | Educational Level | Organized by |
|---|---|---|
| Hiking route | Compulsory secondary | School (IESO Las Villuercas) |
| Geometry on the route | Compulsory secondary | School (IESO Las Villuercas) |
| Study of the mil route | Compulsory secondary | School (IESO Las Villuercas) |
| The water in the geopark | Compulsory secondary | School (IESO Las Villuercas) |
| Gastronomy in the geopark | National diploma | School (IESO Las Villuercas) |
| Gymkhana | Compulsory secondary | School (IESO Las Villuercas) |
| Geopark trails | Compulsory secondary | School (IESO Las Villuercas) |
| Bicycle day | Early childhood and primary | School (IESO Las Villuercas; CEIP Reyes Católicos) |
| Restocking of trees | Primary | School (CEIP Reyes Católicos) |
| Visit to the 14th century hydraulic building | Primary | School (CEIP Reyes Católicos) |
| Chestnut week | Early childhood and primary | School (CEIP Reyes Católicos) |
| The toponymy of the geopark | Sixth orm | High school (IES Mario Roso de Luna) |
| Recycling computer waste | Compulsory secondary and sixth form | High school (IES Mario Roso de Luna) |
| Artistic heritage forgotten in the Puebla de Guadalupe | Compulsory secondary and sixth form | High school (IES Mario Roso de Luna) |
| Haute Provence geological reserve | Sixth form | High school (IES Mario Roso de Luna) |
| Mushrooms in the Villuercas geopark | Compulsory secondary and sixth form | High school (IES Mario Roso de Luna) |
| Minerals and rocks collection | Compulsory secondary | High school (IES Mario Roso de Luna) |
| Discovering the divine proportion of the geopark | Compulsory secondary | High school (IES Mario Roso de Luna) |
| Geopark settlers | Compulsory secondary | Centro escolar (IES Mario Roso de Luna) |
| Geosites | Compulsory secondary | High school (IES Mario Roso de Luna) |
| Geopark subject | Compulsory secondary | High school (IES Mario Roso de Luna) |
| Knowing the geopark | Compulsory secondary | High school (IES Turgalium) |

[1] Colegio de Educación Infantil y Primaria (CEIP)—School of Early Childhood and Primary; [2] Centro Rural Agrupado (CRA)—Rural Center Grouped; [3] Instituto de Educación Secundaria Obligatoria (IESO)—School of Compulsory Secondary; [4] Instituto de Educación Secundaria (IES)—High School; [5] Educación para Adultos (EPA)—Adult Education).

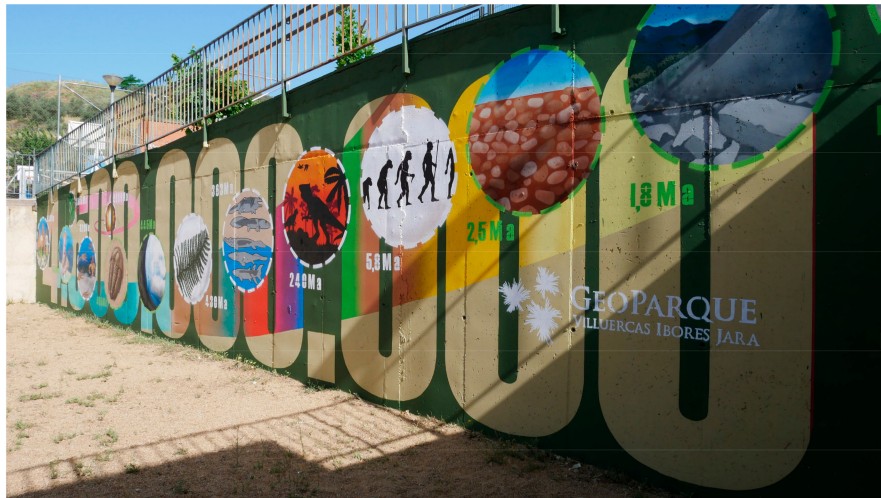

**Figure 5.** Time wall (Fausto Maldonado School). Source: UNESCO Global Geopark Villuercas-Ibores-Jara [56].

All these materials provide a series of educational resources focusing on geology and the environment, targeting the first three stages of schooling (early childhood, primary, and compulsory secondary).

With the common denominator being Las Villuercas Geopark, 81 activities have been organised to cater for primary and secondary education (39.5% primary education; 23.5% compulsory secondary education; 4.9% early childhood education) (see Table 4 and Figure 6). Out of all these, 67.9% are for the sole use of a single stage, as they work on specific aspects of the level they are designed for. The rest are arranged for use in several stages for the purpose of providing continuity in the teaching process (23.5% early childhood–primary; 2.5% primary–compulsory secondary; 3.7% early childhood–primary–compulsory secondary; with the remaining 2.2% being those used on occupational training courses and in adult education). Highlights among all these activities are those of a global nature. These have been created by the geopark's experts and cater for all the pupils and students in the local educational community. The activities of a global nature account for 5.8%, compared to all the other activities (94.2%), which are those created by the schools taking part in the Geoschools project and are actually held at them. The latter are implemented solely at those schools in which they have been designed.

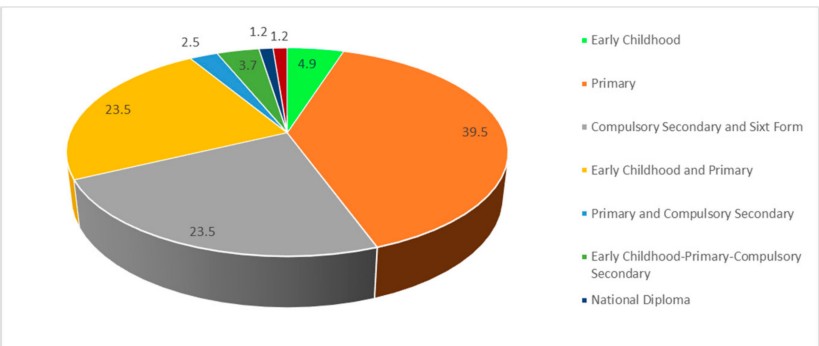

**Figure 6.** Distribution of teaching proposals according to the target stage of education. Source: UNESCO Global Geopark Villuercas-Ibores-Jara and Geopark Text-book. [56,61].

Generally speaking, two large sets of activities have been found according to their subject matter (see Figure 7): those involving nature (79.91%) and those involving a human hand (20.09%). This provides our first finding that confirms the naturalist contribution that a geopark makes to education.

Following this initial interpretation and the grouping of the activities into the set categories, the first results indicate that the most representative type is the one that encompasses those activities that focus on certain features of a territorial nature (46.45%). In other words, these are activities that teach the students about general aspects of the local area (geopark, landscape, geography, etc.). Secondly, those activities centred around geology account for 27.49%. Within this subject matter, and besides rocks and minerals, students learn about different types of reliefs and fossils. The category referring to those activities in which human intervention prevails accounts for 20.09% of the overall total. These mainly focus on traditional economic activities (mining and land uses). As regards conservation and sustainability, these topics are the mainstay of all the other activities (5.97%). Although all the activities have been linked to one or other category depending on their main topic, it should be noted that their teaching/learning process also includes other secondary topics. For example, they all contain an aspect that deals with environmental conservation and geology. Thus, although this is not their main subject matter, they also cater for it. Therefore, in the case of this last category (conservation) in which the percentage is small, it should be noted that those activities being considered are those whose main focus is conservation.

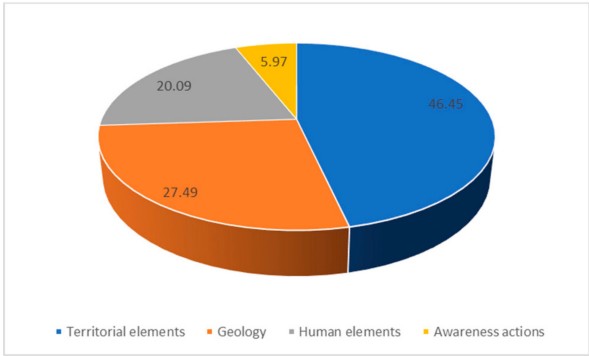

**Figure 7.** Proportion of thematic categories found in the activities. Source: compiled using data provided by the UNESCO Global Geopark Villuercas-Ibores-Jara.

All these activities, besides being classified according to the characteristics of their main topic, have also been analysed on the basis of the keywords that are part of the categories. Accordingly, the keyword that prevails in the description of all of them is geopark (see Figure 8). This aspect helps to confirm that geopark is a key feature in all cases, regardless of the subject matter involved. The second word that appears most often in Spanish is natural. As in the previous case, its presence helps us to stress the importance of natural aspects in them. The same tendency appears both in the subject matter and in the descriptions of the activities: geopark-natural features.

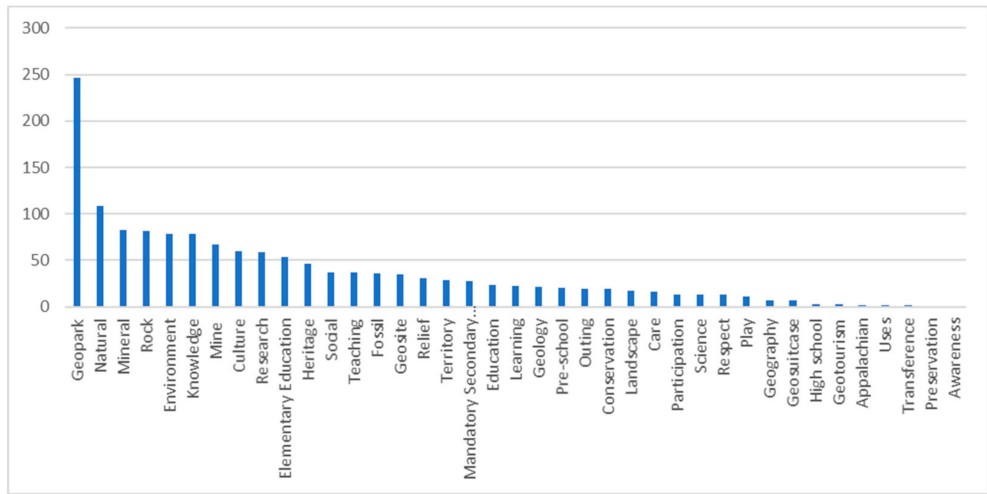

**Figure 8.** Absolute values for each subcategory found in the activities. Source: compiled using data provided by the UNESCO Global Geopark Villuercas-Ibores-Jara [56,61].

In methodological terms, these activities are characterised by the research processes in which, though self-study, the pupils and students search for new information to supplement that provided by their teachers (exploration-based learning process). This means fostering the learning process through discovery and the acquisition of values of respect for the environment and cultural and ecological awareness.

We are dealing with an educational programme, Geoschools, that seeks to include the geopark as a further resource in the teaching-learning process. What's more, this approach is based on a mainstream type of teaching in which self-study based on research is the main method. In general, teaching proposals allow working on content, both in natural sciences and in social sciences, as well as incorporating in some of them certain specific aspects of maths, English, and Spanish. For example, the activity called *Descubriendo la divina proporción el geoparque* [Discovering the divine proportion the geopark] contains content on Maths, Art, Geography and Geology. In turn, the activity of a global nature called *I am a geoparker* has content involving English, geology, biology and geography as its main subjects of focus.

5.2.2. Description and Definition of the Content and Activities in the Textbook Conocimiento del Medio Natural, Social y Cultura del Geoparque Villuercas-Ibores-Jara [Understanding the Natural, Social and Cultural Environment of the Villuercas-Ibores-Jara Geopark].

This supplementary textbook has been created as part of the material resources of the subject that used to be called conocimiento del medio natural, social y cultural (it has now been divided into two new ones: ciencias naturales [natural sciences] and ciencias sociales [social sciences] pursuant to Spain's organic law 8/2013, of 9 December, for improving the quality of education). The following are the school years in which its use is recommended: Years Five and Six in Primary education and years one and two in compulsory secondary education. It is organised around four main pillars: review of prior knowledge; outline of the main content the children are expected to learn; key concepts; and activities. The text is accompanied by illustrations (pictures, graphs and maps) that supplement the written information. This material continues to be used in schools, and now fosters teaching of a mainstream nature by combining two different subjects at the same time. The book contains nine teaching units: one dedicated to a description of the geopark, six dealing with its natural elements and two with human elements. It prompts eight activities in all, in which self-study and investigative processes are the prevailing methods used. All the content studied in the book is backed by examples to be found in the geopark. The following topics are covered: about a geopark; geographical setting of the UNESCO Global Geopark Villuercas-Ibores-Jara; geology, relief; ecosystems; flora; fauna; economic activities;

history, culture and traditions. The teaching units are structured according to the following features: review of prior knowledge through an introductory topic; general theoretical content supported by pictures, maps and graphs; specific theoretical content backed by highlighted points: basic information and concepts on the geopark and its component features, and activities for self-study and group work.

This traditional arrangement of the units informs a teaching method that shifts from the general to the specific, with its point of departure being the pupil's prior knowledge. The knowledge acquired from the units is reinforced by the self-study exercises in which they have to further explore the issues addressed in class by supplementing the content covered.

In sum, the UNESCO Global Geopark Villuercas-Ibores-Jara has a detailed educational project that encompasses the whole of the school community (pupils and teachers) in early childhood, primary and secondary education. Furthermore, the schemes that have been put in place cover the teaching of local natural and cultural aspects, and specifically include a series of proposals in which sustainability is the linchpin of the activities. Besides the programmes arranged for schoolchildren, any visitor may learn for themselves, as the geopark has prepared a series of materials to enable each individual user to understand the basic aspects of the geological or cultural elements at the geosites. This means providing training for society as a whole. As far as teaching is concerned, the geopark is a tool for mainstream teaching. Its use as a core element may lead to the design of sundry topics that make up the study plans. The geopark's educational project has developed activities in social sciences, natural sciences, art, maths, Spanish, and English.

### 5.3. Education in the Geopark: Pupil Involvement

Depending on the characteristics of the activities designed for the children, the geopark's educational project caters for two types of pupils: those enrolled in the Geoschools project and those that come from schools outside the geopark's catchment area. The Geoschools programme involves the entire school population in the area around the geopark. Since 2011, it has recorded an average involvement of 1150 pupils from all levels of schooling (from Early Childhood Education through to the Sixth Form). To date, around 9200 pupils have taken part in this programme. By stages of schooling, the most numerous are the two compulsory ones (see Figure 9), with an average yearly attendance of 450 pupils from primary education and 300 from compulsory secondary education. The average annual figure for both early childhood education and the sixth form is around 200 in both cases, being significantly lower than the two stages mentioned before.

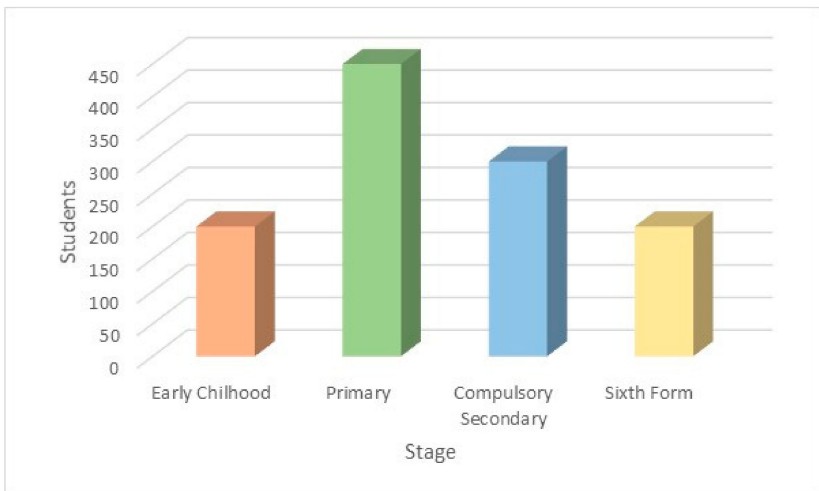

**Figure 9.** Average annual involvement in the Geoschools programme by stage of schooling. Source: compiled using data provided by the UNESCO Global Geopark Villuercas-Ibores-Jara.

This educational programme involves approximately 8.9% of the overall population in the district of Las Villuercas, which has 12,883 inhabitants (Spain's National Office of Statistics—Instituto Nacional de Estadística, June 2019).

Outside the Geoschools programme, but still within the framework of educational collaborations, each year the geopark receives around 100 university undergraduates and master's students from the following universities: Extremadura, and the Autónoma and Complutense in Madrid. The geopark organises activities for these visitors in collaboration with the three universities, and mainly involve fieldwork sessions. This enables the students to supplement their studies in the classroom with work carried out *in situ*.

In addition to the Geoschools programmes and collaborations with universities, the geopark welcomes any school that requests its help in arranging some kind of educational activity within its area. The geopark's range of annual services includes 25 school trips, each catering for around 50 pupils. They involve a day of fieldwork in which the geopark prepares a series of activities to be undertaken. These trips are taken up mainly by schools in Extremadura, whereby approximately 1250 pupils a year learn about the geopark and its characteristics.

The combination of both these approaches leads to an average annual instruction of around 2500 pupils and students from different levels of education. This figure needs to be increased by all those visits that the schools organise independently of the geopark. These trips make the most of the on-line information available on the geosites to organise teaching activities for themselves without the need to arrange a guided tour (in these cases, the geopark does not have any data on the number of visitors involved).

## 6. Discussion

Since it was founded in 2011, the UNESCO Global Geopark Villuercas-Ibores-Jara has consolidated its status as an educational resource for learning about natural and cultural matters within primary and secondary education, and especially in the teaching of geology. Over this time, approximately 20,000 pupils and students have taken part in its educational project, with an annual average of 2500. This means that all the pupils and students within its radius of action and in the immediate vicinity have undertaken one or more of the learning activities and have gone on excursions to the geopark. According to Escorihuela [63] and Głowacki [64] it is important for schools to participate in the educational projects of geoparks in order to supplement the knowledge on Geology. Contact with the environment helps to raise eco-awareness among the population [65], helping to instil values such as respect and favour a significant learning process [66–70]. These aspects are confirmed by the study conducted by Azman et al. [42] in which these authors highlight how instruction in environmental issues within the context of the geopark makes a positive contribution to young people's awareness. In turn, they report that local people with an understanding of the geopark and its resources become guardians of their natural and cultural heritage [42,71,72]. Elsewhere, the studies conducted by Fassoulas and Zouros and Stoffelen stresses the importance of knowledge of the local area for engaging people in the geopark's management processes [10,11], whereby a population that has received a basic education in environmental matters will make a positive contribution to countryside protection and management. Although the educational project in place at the Las Villuercas Geopark is designed mainly for formal education, allowance is made within it for informal training. In this way, moreover, besides those young people that visit within the framework of activities organised by their schools, the geopark receives people interested in learning more about geology. Individual visits and workshops provide people with an understanding that goes beyond what they are expected to learn at school. Thus, the UNESCO Global Geoparks Network becomes a prominent way of fomenting knowledge on geology, in particular, as well as on naturals and cultural elements in the area [73], and of involving the local community in decision-making processes. In addition to these, the study carried out by Farsani et al. [74] entitled Education Program for Drought indicates that knowledge of natural elements helps to understand natural risk. Following this line, UNESCO Global Geopark

Villuercas-Ibores-Jara contributes to the knowledge of natural elements and their possible natural risk through the implementation of workshops

The educational activities organised on Las Villuercas Geopark cater for each and every one of the stages of Spain's formal system of education (early childhood, primary, compulsory secondary, sixth form, and university). Highlights among the range of educational schemes the geopark organises are those that focus upon the stages of compulsory schooling (primary and compulsory secondary education). This ensures that the entire school population in the area of Las Villuercas gains a basic understanding of its natural heritage through the geopark as its common thread. The bulk of the activities can be framed within the official syllabuses, as the content of the subjects of social sciences and natural sciences is related to the subject matter that can be taught based on geoparks. The introduction of geoparks provides a new learning instrument enabling teachers to develop the teaching-learning process in contact with nature and foster geo-education within the ambit of formal education, as posited in the study by Peppoloni and Di Capua [68].

Besides being designed for all stages of formal education, the geopark's educational project provides an extensive range of activities that directly or indirectly cover part of the content of the subjects included in the official curriculum. These activities allow working on curricular content in natural sciences (geology and biology), social sciences (geography, history, and history of art), maths, art and visual education, Spanish, and English. These activities foster the mainstreaming of teaching, as the geopark and contact with the natural environment are used as the basis for working in at least two subjects. Along these lines, we coincide with the research conducted by Ramísio et al. [75] in which they contend that it is possible to raise young people's awareness in environmental matters if this done through contact with nature and by mainstreaming. These authors also note that it is important to use techniques that allow combining the common points across the different subjects for showing pupils that there is a thematic correspondence between these and how their joint and interrelated study facilitates the learning process. One way of adopting a mainstream approach to working on the content of different subjects, and doing so in contact with nature, is through the landscape [76]. Accordingly, geoparks provide landscapes in which the interactions between the natural environment and human activity are obvious and pertinent for explaining different classroom subjects [34]. According to Farsani et al., the landscape is one of the elements, together with geology, that contribute the most to learning about the local area when they do so through geoparks [34]. For instance, the studies undertaken by Swierkosz et al. [77] and Pijet-Migoń and Migoń [78] point out that the presence of abiotic and biotic elements helps to learn about the landscape as well as the relationship between them. Therefore, the more outstanding features in most of the activities involved in the Las Villuercas educational project are the geopark itself, its geology, and its landscape. In addition to its activities, the geopark has prepared a series of teaching materials that directly contribute to the learning processes. The two most widely used are the textbook and the geosuitcase. Not only do they focus on geology, but also on local knowledge, its features and elements, supplementing the instruments and textbooks used in the different schools.

Besides all the teaching options geoparks provide in terms of formal education, they also present economic opportunities [79]. Their twin approach (geotourism and geoeducation) caters for the knowledge transfer required for instructing the general population (informal education) in matters of geology and the environment to enable them to offer a diversified tourist offer and take part in regional management plans [8,10,11,63,80]. To do so, Las Villuercas Geopark has worked within the ambit of informal education through training workshops in which approaches are made to knowledge in matters of geology, the environment, and management and has worked within geotourism through tourism initiatives [81] (museum, mines, etc.). According to Farsani et al. [3] and Ruban [82], geoparks play a role that furthers people's instruction and their engagement with management schemes and plans. This leads to a population that, besides its knowledge of the local area, considers itself part of territorial land planning policies, improving its democratic channels [83]. Accordingly, Las Villuercas Geopark includes local stakeholders (public and private) in its educational processes and also in its

coordination through its Strategic Planning programme that outlines the goals the geopark is to pursue. According to Stoffelen et al. [84], it is essential to train the population to be aware of the value of the territory so that they can use it in a responsible way through Geotourism. UNESCO Global Geoparks Villuercas-Ibores-Jara contributes to this training by creating routes between geosites [85]. Several studies, such as that of Liberatoscioli et al. [86] or Rivero et al [87], have shown that these infrastructures are positively valued by visitors because its help them to learn about geology. Moreover, geoparks need to incorporate new technologies in order to promote the learning-process amongst its visitors. In this sense, mobile applications can be an alternative, as shown in the study carried out by Li et al. [88].

## 7. Conclusions

- UNESCO Global Geopark Villuercas-Ibores-Jara fulfils the two main goals applicable to these kinds of parks: the management, planning and promotion of geotourism, on the one hand, and instruction, education and awareness in environmental matters on the other, with the common link being geology. Its educational project involves all the centres of learning within its catchment area, beginning with early childhood education (preschool) and extending though to higher education. This enables pupils and students to acquire environmental knowledge in contact with nature, thereby increasing their eco-awareness. Furthermore, and through informal education, the geopark seeks to instruct the public at large to provide people with a basic grounding in the local area so that they can suitably take part in local land and territorial management plans. The aim is therefore to mould a society with environmental values that is in a position to develop sustainable tourist initiatives.

- UNESCO Global Geopark Villuercas-Ibores-Jara has developed a solid educational project of which all schools are part. This project is organized for all stages of the Spanish education system. In this, geology is the fundamental part that articulates all project.

- Although all educational stages are present in the educational project, Primary Education has more activities in the educational project.

- Since 2011, 20,000 pupils have participated in educational activities in the geopark. Therefore, the young population can obtain a natural sensibility that in the future can contribute to the fight against climate change.

- The educational project of the geopark provides two educational lines: the one focused on formal education and the one focused on informal education. In this way, the geopark does not only contributes to the training of pupils in its territory, but also to the training of visitors who participate in the activities designed by the geopark for autonomous use.

- All the activities involved in this educational project have the geopark as their common thread, and it serves to orchestrate the different educational disciplines involved. In terms of methodology, the activities combine individual self-study with teamwork, as well as discovery-based learning fieldwork. Furthermore, the activities have been complemented through the creation of a series of educational materials that facilitate the teaching-learning process. The educational process has been designed in cooperation between schools and the geopark so that all the activities can be embedded within the official syllabus and complement the traditional teaching process. This cooperation between teachers and experts at the geopark facilitates the opening of channels for knowledge transfer. Know-how is shared between the schools themselves and between the geopark and the schools. This means that all those involved are aware of the results of the experiences undertaken, and may in coming years choose those that provide the best outcomes.

- The geopark promotes environmental awareness and knowledge of geology through its activities. Seventy-seven activities have been created in collaboration with schools in which the main axis is the geopark.

- The geopark contributes to the formation of citizenship in environmental and geological issues so that they can participate in the elaboration of the land management proposals. To do this, they

have created territorial analysis workshops. Since 2011, an annual workshop has been developed to raise public awareness of the importance of protecting natural resources.

- The activities developed in the geopark contribute to the promotion of collaborative work and to mainstreaming in the teaching processes.
- From this first analysis on the educational characteristics proposed by the geopark, two new lines of work must be carried out: first, a comparative analysis between the educational projects of several geoparks. Secondly, an analysis of the results obtained by the students after the development of these activities.

**Funding:** This research received founds from Research Group Observatorio de la Marca España en Europa (OMEU) hosted at the University of Salamanca, Spain.

**Acknowledgments:** I should like to thank the UNESCO Global Geopark Villuercas-Ibores-Jara for the information provided on its facilities and the activities it undertakes.

**Conflicts of Interest:** The author declare no conflict of interest.

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
