# Peer review of "Geoparks and Education: UNESCO Global Geopark Villuercas-Ibores-Jara as a Case Study in Spain"

_geosciences, doi:10.3390/geosciences10010027_

Round 1
Reviewer 1 Report
This paper offers in-depth analysis of the educational potential of one UNESCO geopark, which seems to be representative example on the international scale. The paper is informative and contributes to geopark conceptualization. I recommend its publication after certain improvements – see suggestions below.
1) Some important literature is missed. First, you MUST cite these works: Henriques, M.H., Brilha, J., 2017. UNESCO Global Geoparks: a strategy towards global understanding and sustainability. Episodes 40, 349-355; Dowling, R., 2011. Geotourism's Global Growth. Geoheritage 3, 1-13; Dowling, R., Newsome, D. (Eds.), 2018. Handbook of Geotourism. Edward Elgar, Cheltenham, 499 pp. Second, try to collect more citations to papers addressing the educational role of geotourism (e.g., see papers in Geosciences, Resources, e-Review of Tourism Research, etc.). Third, you need to cite all papers devoted to the analyzed geopark – I have detected several papers in Scopus by various specialists: feel free to select which one(s) to cite, but I'd prefer citations to all of them – work with Scopus, please. Moreover, in Discussion you should relate your own findings to the information from the other papers in this geopark.
2) Why captions are above the figures? Unusual and incorrect!
3) Figure 1: what do the numbers mean? Geosites? These should be named!
4) The existing section 4 should be split into three separate sections, namely Geopark setting (this should occur before Methodology, and it should contain geographical and geological sub-sections), Results (your own findings), and Discussion (interpretation of your findings, putting these findings into the global context, and practical recommendations – some information from the existing section Discussion and Conclusion should be used there).
5) The last section should be entitled Conclusion, and it should be the numbered list of the main findings and recommendations for further studies.
6) The readers would be interested in seeing geological map with all main localities.
7) Why not to provide 1-3 photos of the educational infrastructure/educational events in this geopark?
Author Response
RESPONSES TO REVIEWER 1
1. Some important literature is missed. First, you MUST cite these works: Henriques, M.H., Brilha, J., 2017. UNESCO Global Geoparks: a strategy towards global understanding and sustainability. Episodes 40, 349-355; Dowling, R., 2011. Geotourism's Global Growth. Geoheritage 3, 1-13; Dowling, R., Newsome, D. (Eds.), 2018. Handbook of Geotourism. Edward Elgar, Cheltenham, 499 pp. Second, try to collect more citations to papers addressing the educational role of geotourism (e.g., see papers in Geosciences, Resources, e-Review of Tourism Research, etc.). Third, you need to cite all papers devoted to the analyzed geopark – I have detected several papers in Scopus by various specialists: feel free to select which one(s) to cite, but I'd prefer citations to all of them – work with Scopus, please. Moreover, in Discussion you should relate your own findings to the information from the other papers in this geopark .
Done. I have added four new bibliographical references.
2. Why captions are above the figures? Unusual and incorrect!
Done. I have named the figures according to the rules for the authors.
3. Figure 1: what do the numbers mean? Geosites? These should be named!
Done. I have added a table with the geosites
4. The existing section 4 should be split into three separate sections, namely Geopark setting (this should occur before Methodology, and it should contain geographical and geological sub-sections), Results (your own findings), and Discussion (interpretation of your findings, putting these findings into the global context, and practical recommendations – some information from the existing section Discussion and Conclusion should be used there).
Done. I have made the proposed changes and a new restructuring
5. The last section should be entitled Conclusion, and it should be the numbered list of the main findings and recommendations for further studies.
Done.
6. The readers would be interested in seeing geological map with all main localities.
Done
Why not to provide 1-3 photos of the educational infrastructure/educational events in this geopark?
Done
Reviewer 2 Report
The manuscript in general is written in very good English and is well structured focusing on the topic of geo-education in UNESCO Global geoparks, with Villuercas-Ibores-Jara as a case study. Author used an appropriate methodology to collect and study data and the results sound very interesting and well-documented. International literature was properly and extensively used, while charts and graphs are clear and well-illustrated (Fig. 2 appears very dark and needs improvement).
Apart from some very minor spelling and grammar comments as well as the use of some terms such as instruction (that probably refers to training) and learning methodologies (experiential, exploration-based are terms that could have been used), I would like to add two minor comments.
The description of the situation, role and structure of the UNESCO Global Geoparks, as well as the Global and Regional networks should be more carefully and accurately presented (some comments exist in text). To further support and strengthen the manuscript’s results, data related to the assessment of the applicability and the effectiveness of the educational products by the end users (i.e. pupils, teachers, individuals, visitors), would be very useful. If such already exist would be good to incorporate.

Author Response
Comment and Suggestions for Author made by the reviewer:
1- Done. I have removed somo aspects to be more precise.
2- At the momento I cannot provide results on the evaluation of the students because it will be the second phase of the study.
3- Comments in the enclosed file: done.
Thank you very much for your comments and suggestions.
Reviewer 3 Report
My general impression is that this paper reports facts and activities rather than sets any scientific problem and consequently solves it. As a result, there are no points where I can argue with the author as all is based on evidence. However, in terms of originality and novelty, I gained little, apart from learning that nature is exposed in educational activities offered in the Geopark, including by the Geopark. Given what Geoparks are created for, this is nothing unexpected. I see the scope to enlarge discussion by looking at other geoparks (maybe just those in the Iberian Peninsula) to see whether their educational programmes are similar or not, and why. I also wonder if "Geosciences" is the right outlet for this paper. To me, it fits much more into the field of Geographical Education rather than Geosciences as such.
Having said that and acknowledging that the Editors will make final decision, I recommend the following action at the revision stage:
The entire Introduction (L24-103) is overlong and includes issues not really directly relevant to this paper. It can be easily summarized into several key points which provide the right context. 20-30 lines should be more than enough to achieve this.
I disagree with the statement (L113-114) that geotourism "converts into a type of tourism in which cultural aspects are one of the principal enticements". Geotourism remains focused on geology and physical landscapes, otherwise it will have no meaning and will be operationally indistinguishable from other "tourisms". What the author talks about in L114-116, is recognition that very few tourists will have narrow interests in geology only and they expect a more balanced, holistic offer. But this is something different that defining Geotourism.
As above, text in L112-164 can be considerably shortened, without detrimental effect on the message. All this can be reduced to a few lines to say that geoeducational activities are among the core activities in Geoparks.
More information is needed how subcategories were selected, on what basis (L188, Table 1).
Instead of an overlong introduction (see comments above) the reader will be more interested to learn more about the nature of geosites in the Geopark: what they show? which subjects? how they are prepared for tourists etc. A few representative photographs would enhance the paper.
L377-379 How general aspects (geography, landscapes) were separated from more specific geology, if the latter contains "relief"? Relief is exactly landscape (actually, the most important aspect of it) and definitely part of geographical characterization of an area.
Fig. 6 does not show "proportions" (as in Caption) but absolute values.
Logic of construction of Fig. 7 is unclear and I am not sure if this figure is necessary at all. What extra message it conveys?
Author Response
1. The entire Introduction (L24-103) is overlong and includes issues not really directly relevant to this paper. It can be easily summarized into several key points which provide the right context. 20-30 lines should be more than enough to achieve this.
I consider that I should not reduce the introduction because it is structured according to the general lines of geoparks: geotourism and geo-education. I begin with a brief historical approximation of geoparks to explain the two fundamental axes. It is necessary to understand the structure of work.
2.I disagree with the statement (L113-114) that geotourism "converts into a type of tourism in which cultural aspects are one of the principal enticements". Geotourism remains focused on geology and physical landscapes, otherwise it will have no meaning and will be operationally indistinguishable from other "tourisms". What the author talks about in L114-116, is recognition that very few tourists will have narrow interests in geology only and they expect a more balanced, holistic offer. But this is something different that defining Geotourism.
Partially modified. I have changed the statement related to cultural aspects to indicate that culture is part of geotourism. I have not modified the rest because the geotourism proposed by geoparks is based on a global tourism in which all the elements offered by the territory are used. Although it is true that geology and relief are the most prominent.
As above, text in L112-164 can be considerably shortened, without detrimental effect on the message. All this can be reduced to a few lines to say that geoeducational activities are among the core activities in Geoparks.
In this part it is preferable to make a deep approach to geo-education
More information is needed how subcategories were selected, on what basis (L188, Table 1).
Done.
Instead of an overlong introduction (see comments above) the reader will be more interested to learn more about the nature of geosites in the Geopark: what they show? which subjects? how they are prepared for tourists etc. A few representative photographs would enhance the paper.
This suggestion is very good and will be taken into consideration for future work focused on teaching practice. In this case I have only listed the geosites because I needed to focus on educational activities.
Fig. 6 does not show "proportions" (as in Caption) but absolute values.
Done
Logic of construction of Fig. 7 is unclear and I am not sure if this figure is necessary at all. What extra message it conveys?
This figure (previously nº7 and currently nº 8) represents the number of times that subcatagetries are located in the description of the educational activities of geoparks. This is intended to highlight the importance of the geopark in educational activities.

Round 2
Reviewer 1 Report
I'm generally satisfied with this revision. The author did a great job, and the paper looks much better now. It deserves publication after MINOR revision. I do not wish to be over-critical, but I still see three important, although very easy-to-address issues.
1) The number of literature is still insufficient. I re-type of a part of my recommendation from the original review. Please, try to collect more citations to papers addressing the educational role of geotourism (e.g., see papers in Geosciences, Resources, e-Review of Tourism Research, etc.). ATTENTION: you need to cite all papers devoted to the analyzed geopark – I have detected several papers in Scopus by various specialists: feel free to select which one(s) to cite, but I'd prefer citations to all of them – work with Scopus, please. Moreover, in Discussion you should relate your own findings to the information from the other papers in this geopark .
2) In the caption of Figure 3, it should be noted that the numbers are explained in Table 1.
3) The language still needs some polishing.
Author Response
1-The number of literature is still insufficient. I re-type of a part of my recommendation from the original review. Please, try to collect more citations to papers addressing the educational role of geotourism (e.g., see papers in Geosciences, Resources, e-Review of Tourism Research, etc.). ATTENTION: you need to cite all papers devoted to the analyzed geopark – I have detected several papers in Scopus by various specialists: feel free to select which one(s) to cite, but I'd prefer citations to all of them – work with Scopus, please. Moreover, in Discussion you should relate your own findings to the information from the other papers in this geopark
1- Done
2- In the caption of Figure 3, it should be noted that the numbers are explained in Table 1.
2- Done
3- The language still needs some polishing.
3-Done
